# Synthesis of Metallic Nanostructures Using Ionizing Radiation and Their Applications

**DOI:** 10.3390/ma17020364

**Published:** 2024-01-11

**Authors:** Hynd Remita, Isabelle Lampre

**Affiliations:** Université Paris-Saclay, CNRS, Institut de Chimie Physique, UMR 8000, 91405 Orsay, France; hynd.remita@universite-paris-saclay.fr

**Keywords:** ionizing radiation, radiolysis, metallic nanoparticles, nucleation, redox properties, optical properties, catalysis, energy, radiotherapy

## Abstract

This paper reviews the radiation-induced synthesis of metallic nanostructures and their applications. Radiolysis is a powerful method for synthesizing metallic nanoparticles in solution and heterogeneous media, and it is a clean alternative to other existing physical, chemical, and physicochemical methods. By varying parameters such as the absorbed dose, dose rate, concentrations of metallic precursors, and nature of stabilizing agents, it is possible to control the size, shape, and morphology (alloy, core-shell, etc.) of the nanostructures and, consequently, their properties. Therefore, the as-synthesized nanoparticles have many potential applications in biology, medicine, (photo)catalysis, or energy conversion.

## 1. Introduction

The use of ionizing radiation, i.e., X-rays, gamma-rays, and particles beams, is well-known in the medical field for diagnosis (radiography, scintigraphy) and therapy (radiotherapy, hadrontherapy) [1,2]. But, ionizing radiation has many other beneficial applications in various fields. Industrial irradiation facilities constitute a growing market, with approximately 300 gamma irradiators and electron accelerators around the world [3]. Their numerous applications include the preservation of cultural heritage, sterilization of drugs and health care products, modification of polymers, and depollution [4]. In this review, we tackle the use of ionizing radiation in the nanoscience domain.

Nanoscience is a research area of immense interest because of the various applications of nanoparticles (NP) in a large number of fields, including catalysis, optics, electronics, and medicine. Indeed, nanoparticles have attracted much attention in recent decades because of their unique properties, which differ from those of their bulk counterparts [5,6,7]. Many of their properties are size-dependent due to their high surface-to-bulk atom ratio and are influenced by their morphology and affected by their environment. In this paper, we will focus on metallic nanostructures (MNS), and in particular, on those formed from noble metals, as they offer a large variety of properties (optical, catalytic, antimicrobial, etc.) [8,9]. For example, gold, which is inert in its bulk solid state, acquires remarkable catalytic properties on a nanometric scale [10,11]: small gold nanoparticles can oxidize carbon monoxide (which is highly toxic) into carbon dioxide at room temperature [12]. Gold also changes color at the nanometric scale, with spherical gold nanoparticles appearing red in solution (their deposition on a support such as silica or alumina induces a red or pink color of the powder, depending on the content) [13,14]. The size and shape dependence of the properties of nanoparticles has induced several physical and chemical methods of synthesis, which are commonly gathered in two approaches: the bottom-up approach (from atoms or molecules) and the top-down approach (from bulk materials) (Figure 1) [15]. Among the bottom-up methods, radiolytic synthesis using ionizing radiation is a powerful method for producing metallic nanoparticles in solution and heterogeneous media, which offers many advantages compared to other techniques [16,17,18,19].

This review first presents an overview of ionizing radiation and its radiolytic effects used for the synthesis of MNS. Then, the main principle and processes involved in the radiolytic synthesis of MNS in protic solvents are described, as well as the effects of different parameters, such as the dose rate and the use of stabilizers. Finally, examples of the properties and applications of the formed nanostructures are given.

## 2. Ionizing Radiation and Radiolysis

Ionizing radiation can be either an electromagnetic wave or a particle beam that carries enough energy to be able to remove a tightly bound electron from the orbit of an atom (core electron), causing the atom or molecule to become charged (ionized). The ejected electron, called the secondary electron, may have sufficient energy to cause further ionizations. By interacting with matter, the ionizing radiation loses energy, which is absorbed by the matter. The energy absorbed by the material, called the dose, is expressed in grays (1 Gy = 1 J kg^−1^), and the amount of energy absorbed per unit of time, the dose rate, is expressed in Gy s^−1^. 

### 2.1. Electromagnetic Waves or High-Energy Photons

X-rays (energy range of 0.1 to 200 keV) and gamma-rays (energy range of 10 keV to 10 MeV) are considered to be ionizing radiation. The distinction between X-rays and gamma-rays comes from their origin. X-rays are emitted from the rearrangement of the electrons of a highly excited atom, while gamma-rays originate from the rearrangement of the nucleus in a nuclear reaction. Even if ultraviolet radiation can ionize matter, in particular far UV, it is not considered to be ionizing radiation, as it ejects only valence electrons and is related to photochemistry. 

X-rays and gamma-rays have similar properties and interact similarly with matter according to three main interactions: the photoelectric effect, the Compton effect, and the pair production. The predominant range of these effects depends on the energy of the incident photon and the atomic number, Z, of the atom with which it interacts. Consequently, for X-rays or γ-rays interacting with matter, a linear attenuation coefficient, which depends on the photon energy and nature of the matter, is defined. This coefficient enables the determination of the energy deposited in a given thickness of matter, quantified as “Linear Energy Transfer” or LET. Photonic radiation is considered to be low-LET radiation but it can penetrate quite deeply inside matter.

The main gamma-sources are based on ^60^Co or ^137^Cs isotopes and are used in a continuous radiation mode. ^60^Co sources emit photons having energies of 1.17 and 1.33 MeV, and ^137^Cs sources deliver photons of 0.66 MeV. X-ray production comes from particle accelerators and can be used as continuous or pulsed sources. 

### 2.2. Particle Beams

Accelerated particle beams of electrons or heavy ions (nuclei) constitute the other kind of ionizing radiation used to synthesize MNS. In contrast to photons, electrons and heavy ions have a mass and a charge, so they interact with matter mainly through electrostatic interactions. For both, the main interaction consists of inelastic collisions or scattering (although for electrons, the Bremsstralhlung process and Cerenkov effect must also be considered). However, due to the large difference in mass between electrons and heavy ions (for instance, the mass of a proton is 1836 times the mass of an electron), the loss of energy by collision differs significantly; accelerated electron beams are low-LET ionizing radiation (similar as γ-rays) while heavy ions beams are high-LET radiation. Consequently, in the case of electrons, the LET does not depend so much on the electron energy. In contrast, for heavy ions, the LET changes significantly with the particle energy when the ion is slowed down inside matter and most of the energy is lost at a specific penetration distance called the Bragg peak.

Different types of accelerators exist to produce particle beams; cyclotrons and linear accelerators are the most common ones. Electron beams are by far the most common beams used, in particular in pulse mode for kinetic studies. Accelerated electrons usually have energies in the range of 2–10 MeV. 

### 2.3. Radiolysis

Radiolysis gathers all the events induced by ionizing radiation that transform the irradiated medium; it is quite complex as ionizing radiation induces ionization and excitation, and subsequently, dissociation, decomposition of molecules, and formation of new species. These events depend on the total energy deposited by the ionizing radiation, i.e., the dose, but also on the way the energy is deposited (low LET or high LET) and the rate at which it is deposited (high dose rate or low dose rate). Typically, γ-rays sources have low dose rates (below a few Gy/s) while accelerated particle beams interact with high dose rates (typically several kGy/s for electron beams).

It must be noted that ionizing radiation interacts non-selectively with the medium, so atoms or molecules are ionized according to their abundance in the medium and their electronic density. In particular, for diluted solutions, i.e., with solute concentrations below 0.1 mol L^−1^, the ionizing radiation interacts mainly with the solvent molecules.

The irradiation of a polar protic medium by ionizing radiation induces mainly the excitation and ionization of the solvent molecules, RH (here, R represents OH for water, NH_2_ for liquid ammonia, R’O with R’ alkyl or aryl for alcohols, etc.). The excited molecules, RH*, dissociate to give hydrogen atoms H^•^ and radicals R^•^. Upon ionization, the ejected electrons progressively lose their kinetic energy, in turn causing excitation and ionization of the solvent molecules. When their kinetic energy reaches the same order of magnitude as the thermal energy, the electrons become thermalized and solvated (e^−^_s_). The primary cations formed during ionization undergo ion–molecule reactions, giving rise to radicals. These primary species, formed in close proximity, can react together (dimerize, recombine), leading to molecular species, and diffuse. After approximately a hundred nanoseconds, all of these species are homogeneously distributed in the medium. They can further react together or with present solutes [20]. At the end of the primary stages characterized by non-homogeneous kinetics (Figure 2), the radiolysis of a protic solvent RH can be written: RH ⇝ R^•^, H^•^, e^−^_s_, RH_2_^+^, R_2_, H_2_(1)

The amounts of formed species depend on the solvent and the ionizing radiation (mainly through the LET) and are given by the radiation chemical yields, G, in mol/J (number of species produced for a dose of 1 J). Table 1 gives the G-values of the primary species formed in water for γ-rays and alpha (He^2+^) particles. The radiolytic yield and concentration C of a species are linked by the relation:C = ρ D G(2)
where D is the absorbed dose and ρ the density of the solution. 

## 3. Synthesis of Metallic Nanostructures Using Ionizing Radiation

### 3.1. Principle and Advantages

Radiolytic synthesis of metallic nanostructures (MNS) consists of the irradiation by ionizing radiation of a solution of a polar protic solvent containing metallic ions or complexes in the presence of ligands, polymers, surfactants, or supports to limit the growth of the produced nanoparticles [16,18,19,22]. This method benefits from the radiation–matter interactions generating the reducing species and from the combination of reduction and aggregation mechanisms (Figure 3).

The main advantages of the radiolytic synthesis of MNS are (i) synthesis at ambient temperature and atmospheric pressure, (ii) homogeneous reduction and nucleation leading to a narrow size distribution of particles, (iii) no chemical reducing agents added as the reducing species come from the solvent, (iv) a very high reducing power of the produced solvated electrons, (v) the ability to control the morphology and structure of bimetallic nanostructures with the dose rate, (vi) the possibility of synthesis in situ, on or inside supports and in complex media, and (vii) the study of the first reduction and nucleation steps by pulsed radiolysis. 

However, to take advantage of these benefits, it is necessary to be aware of the limitations of this method and, in particular, the depth of penetration of the ionizing radiation in the matter. While X-rays and γ-rays can penetrate matter over great distances (few tens of centimeters in liquid water), electron beams are stopped very rapidly (a few cm in water). The presence of solutes at high concentrations may affect the absorption of the radiation by the solution. Thus, control of the concentrations or application of agitation to increase uniformity in exposure to radiation may be necessary. In industrial facilities, the homogeneity of gamma-irradiation is ensured by rotating the packages in front of the source. 

### 3.2. Radiolysis of Solvent and Reduction of Metallic Ions

Ionizing radiation sent into a solution containing metallic ions or complexes in a polar protic solvent RH excites and ionizes the most abundant molecules, those of the solvent. After the primary non-homogeneous processes (Figure 2), the formed species (Equation (1)) have diffused and are homogeneously distributed in the solvent; they may react with the solutes.

Among the produced species, the solvated electron, e^−^_s_, and the hydrogen atom, H^•^, are strong reducing species which have very negative redox potentials (for instance in water E°(H_2_O/e^−^_aq_) = −2.87 V_SHE_ and E°(H_3_O^+^/H^•^) = −2.3 V_SHE_ [23]); thus, these species can easily reduce, even at room temperature, any metallic ions in solution. According to the RH solvent, the R^•^ radical can either be reducing (for instance, R’O^•^ radicals for alcohols) or oxidative (for instance, ^•^OH radicals for water). In aqueous solution, the hydroxyl radical ^•^OH has a strong oxidative potential (E°(^•^OH/H_2_O) = 2.8 V_SHE_ [23]) and is capable of bringing metallic ions or atoms to a higher oxidation state. To avoid oxidation, ^•^OH scavengers such as secondary alcohols or formate ions are usually added to the solutions; these compounds react with ^•^OH and form reducing radicals (Table 2) which can also contribute to the reduction of the metallic ions. Ter-butanol (2-methyl propan-2-ol) is also frequently used to react with ^•^OH, but in that case, the formed radical is of low reactivity and does not react with metallic ions.

Thus, the solvated electrons and the reducing radicals can reduce the metallic ions M^+^ or M^Z+^ via successive reactions:M^Z+^ + e^−^_s_ (or H^•^ or R^•^) → M^(Z−1)+^
(3)
⋮
M^+^ + e^−^_s_ (or H^•^ or R^•^) → M^0^(4)

It must be noted that, to have a favorable reducing environment, the presence of dissolved dioxygen must be removed either by freeze–pump–thaw degassing or by bubbling with inert gases (N_2_ or Ar) and such conditions must be maintained during irradiation.

### 3.3. Nucleation and Growth of Clusters

The interactions of ionizing radiation with the medium induce a homogeneous distribution of the reducing species and, consequently, a homogeneous distribution of the first metallic atoms, initiators of the nanostructures. As the binding energies between metallic atoms or between metallic atoms and ions are higher than the binding energies with the solvent molecules or ligands, the metallic atoms and ions associate when they encounter one another. First, dimers are formed (Equations (5) and (6)), and then with further associations and coalescence (Equations (7)–(10)), clusters, aggregates, and nanostructures are progressively produced (Figure 2):M^0^ + M^0^ → M_2_(5)
M^0^ + M^+^ → M_2_^+^(6)
M_2_^+^ + M_2_^+^ → M_4_^2+^(7)
M_m_ + M^+^ → M_m+1_^+^(8)
M_i_^x+^ + M_j_^y+^ → M_i+j_^(x+y)+^(9)
M_m_ + M_p_ → M_n_(10)
where the subscripts i, j, m, p, and n indicate the nuclearity of the aggregates, i.e., the number of associated metallic nuclei (in the form of atoms or ions) and the superscripts x and y indicate the number of charges related to the presence of ions. 

While associations proceed, reduction processes also occur on oligomers, consuming the reducing species without generating new free atom initiators of a new particle.

Pulsed radiolysis enables to study the initial reduction of metallic ions, the association between ions and atoms, their reduction, and the coalescence mechanisms. For instance, the reduction of silver ions, Ag^+^, alone or complexed with different ligands was studied in various solvents, and the first transient species (in particular, the silver atom Ag^0^ and the charged dimer Ag_2_^+^) were characterized by their optical spectra [25,26]. In the case of gold, it was shown that the trivalent gold complex [Au^III^Cl_3_]^−^ is reduced in water into a divalent complex, which disproportionates into mono and trivalent ions [27].

Using pulsed radiolysis, it was also shown that the redox properties of metals change when they are atoms or aggregates of a few atoms. Indeed, the redox potential of the (M^+^_(solv)_/M^0^_(solv)_) couple where M^+^_(solv)_ and M^0^_(solv)_ are the solvated ion and atom, respectively, differ from the redox potential of the (M^+^_(solv)_/M_(s)_) couple where M_(s)_ corresponds to the bulk metal. For instance, in aqueous solution, for silver the redox potential of *E°*(Ag^+^_(aq)_/Ag^0^_(aq)_) was determined to be equal to −1.8 V_SHE_ while *E°*(Ag^+^_(aq)_/Ag_(s)_) = 0.79 V_SHE_; similarly, for copper, *E°*(Cu^+^_(aq)_/Cu^0^_(aq)_) was estimated to be –2.7 V_SHE_ while *E°*(Cu^+^_(aq)_/Cu_(s)_) = 0.52 V_SHE_ [28]. It is also worth noting that the redox potential depends on the presence of ligands and complexation (Table 3). 

Despite their short lifetime during the growth, it was established that the redox potential of the aggregates (M_n_^+^/M_n_) increases with the nuclearity n (Figure 4); thus, it becomes easier to reduce a metallic ion on the surface of a larger aggregate [32]. Reductants (other than the solvated electrons) with a redox potential too high to reduce the isolated atoms can reduce the metallic ions adsorbed on the surface of aggregates already formed by radiolysis and contribute to the growth of the nanoparticles. 

As the reducing species are homogeneously produced in the medium, this is also the case of the first atoms and clusters; subsequently, associations and coalescence lead to a similar growth of the aggregates resulting in monodisperse nanoparticles. 

### 3.4. Dose Rate Effects on the Size and Morphology of the Nanostructures

The reduction of metallic ions down to the zero-valent atoms depends on the total amount of reducing species produced by ionizing radiation, i.e., the dose or deposited energy in the medium. However, the reduction reactions occur in parallel with the association and coalescence phenomena, so they may concern free ions in solution or ions already agglomerated. Consequently, the rate to produce the reducing species, i.e., the amount of reducing species present at the same time, plays a role in the growth of the aggregates and the size of the final nanostructures. The rate of formation of reducing species is linked to the rate of energy deposition, the dose rate (Figure 5).

At a high dose rate (typically higher than a few tens of Gy/s, but that depends on the metallic precursors and the medium), high amounts of reducing species are produced in short times. Hence, the reduction of ions is fast, and numerous free atoms are produced simultaneously. Each atom is potentially a nucleation center, a start-point (seed) for the growth of a nanoparticle. Consequently, many aggregates or particles are formed, with a small final size and the distribution in size of the nanoparticles is narrow (Figure 5a).

At a low dose rate (lower than a few Gy/s), few reducing species are formed simultaneously, so the reduction rate is slow compared to the association processes. At the beginning of irradiation, few free atoms are produced and, the reducing species react with already-formed oligomers. Thus, as the number of nucleation centers or seeds is small, the number of aggregates and formed particles will be smaller compared to that produced at a high dose rate, and the final size of the nanoparticles will be larger for the same initial concentrations of metallic ions (Figure 5b). Due to the increase in the redox potential of the aggregates with their size, the reduction of ions on the surface may also occur by reactions with reductants (electron donor D) which cannot reduce the free ions; these chemical reductions also contribute to the growth of the nanoparticles (Figure 5c) [34].

In the case of the synthesis of bimetallic nanoparticles, the dose rate may also affect the structure of the formed nanoparticles: core-shell or alloy. The obtained structures result from kinetic competition between reduction, association, and intermetallic electron transfer processes. Considering an equimolar mixture of two metallic ions, M^+^ and M′^+^, the probabilities that the radiolytic reducing species react with either M^+^ or M′^+^ are quite similar, but when atoms and ions associate, an electron transfer may occur from the less noble metal (for instance, M′) to the more noble metal (M here) leading to the reduction of the more noble metal:M^+^ + M′ → (M^+^M′) → (MM′^+^) → M + M′^+^(11)

Consequently, ions of the less noble metal act as electron relays during the reduction; aggregates of the M metal form first, and when all M^+^ ions are reduced, M′^+^ ions are then reduced on the surface of the M_n_ aggregates; thus, the produced nanoparticles have a core-shell structure with the more noble metal in the core. In the case of multivalent ions, electron transfers between the low valences of the ions are also possible, favoring the probability of segregation of the two metals. The intermetallic electron transfer rate depends on the two metals [35]. If this transfer is not too fast, it may be avoided by using ionizing radiation with a high dose rate. At a high dose rate, a large amount of M and M′ atoms are generated within a very short time, which favors the combined coalescence of M and M′ atoms and the formation of alloys (Figure 6) [36]. It must be noted that high dose rates induce the formation of alloyed nanostructures that have the same metal ratio as their ionic precursors. 

This dose rate effect on the nanoparticle structure was first studied with the formation of bimetallic Au-Ag nanoparticles from aqueous solutions containing both KAuCl_4_ and Ag_2_SO_4_ salts [37]. The irradiation of the solutions by γ-rays with a low dose rate (1.06 Gy/s) leads to the formation of core-shell nanoparticles with gold, the more noble metal, in the core. In contrast, irradiation by γ-rays or electron beams with high dose rates (9.7 Gy/s and 2194 Gy/s, respectively) results in the production of alloyed Au-Ag nanoparticles (where silver and gold atoms are homogeneously distributed in the nanoparticles). This dose rate effect was further studied and used to synthesize different bi-metallic nanoparticles, for instance, Au-Pt [38,39] and Au-Pd [40]. Several Ni-based nanoparticles were also synthesized by room-temperature radiolysis [41,42]. It was shown that different stoichiometries of Ag-Ni alloy NP and Pd_0.5_-Ni_0.5_ alloy NP can be synthesized using the high-dose-rate radiolytic method [41]. Reference [43] provides a review of the nanoparticle alloy formation by radiolysis. 

It should be noted that, for otherwise similar conditions, the size and structure of nanoparticles generated by ionizing radiation are mainly controlled by the dose rate. Nevertheless, the TEL strongly modifies formation yields and slightly influences nanoparticle shape [39].

### 3.5. Stabilization of the Formed Nanostructures and Supported Nanoparticles

The size of the nanostructures produced by ionizing radiation can be affected by the dose rate, but to produce stable nanoparticles with a well-controlled size and avoid complete coalescence and agglomeration of the particles, the use of stabilizers or supports is often necessary. 

Different kinds of stabilizers or capping agents (ligands, surfactants, or polymers) can be added to the solutions to coat the synthesized nanostructures and prevent their agglomeration via steric and/or electrostatic repulsion interactions. However, attention must be paid to the concentrations and redox properties of the chosen species, as they may influence the growth of the particles by chemical reduction of the metallic ions adsorbed on the surface (Figure 4). Polymers constitute the main class of stabilizers used in radiolytic synthesis: polyvinyl alcohol (PVA) [39,44,45,46], polyvinyl pyrrolidone (PVP) [47,48,49], and polymethylmethacrylate (PMMA) [50] are among the most frequently used. Natural polymers such as chitosan or polypeptides were also used to stabilize metal nanoparticles synthesized by ionizing radiation [51,52]. Surfactants such as cetyltriethyl ammonium bromide, CTAB, and sodium dodecyl sulfate, SDS, are also used [53,54,55,56,57]. In general, these stabilizers possess functional groups which have a good affinity with the metal allowing the coating of the metallic aggregates and nanostructures, and then avoid the coalescence by steric hindrance, sometimes also coupled with electrostatic repulsion. Lately, organic macrocycles, cyclodextrins, or calixarenes have appeared as interesting stabilizers due to their conformational behavior and chemical versatility. Indeed, ring functionalization may induce a good anchoring on the surface of the nanoparticles while maintaining good accessibility, which is a key factor in catalysis. α-cyclodextrin-coated Pd nanoparticles with different morphologies were produced by electron beams and γ-rays and were shown to have catalytic and anticancer properties [58]. Mono- and bi-metallic gold–silver nanoparticles stabilized by calix[8]arenes were synthesized by radiolysis and showed catalytic properties [59]. Small ligands, such as cyanide ions, CN^−^, can stabilize small nanoparticles by electrostatic repulsions without the addition of macromolecules, due to their strong interactions with the metallic structures. Such interactions may also change the redox properties, as it was evidenced that, at a given nuclearity, silver clusters were more noble than gold clusters in a cyanide environment [60]. Under a CO atmosphere, the radiolytic reduction of platinum salts in alcohol (ethanol or 2-propanol) solutions leads to the formation of small Chini clusters, [Pt_3_(CO)_6_]_n_^2−^, and the number n of triangular arrangement (n = 3 to 10) depends on the dose [61].

Another way to control the size and shape of the formed nanostructures is to use micellar solutions [62] or mesophases (Figure 7). For example, combining the self-assembly properties of amphiphilic molecules with radiolysis, core-shell structures consisting of nanometric linoleate spherical micelles as the core and silver as the shell were synthesized [63]. A one-pot synthesis of gold nanorods (Au NRs) with a controlled aspect ratio (ratio of length over diameter) was performed by radiolysis using CTAB as a surfactant, while similar syntheses require two steps by a chemical method [64,65]. These nanorods can be embedded in PVA gels formed by crosslinking of the polymer under irradiation [64]. Mesophases or liquid crystal phases offer different structures (lamellar, hexagonal, cubic) and constitute interesting media as they may serve as soft templates to generate 1D, 2D, or 3D nanostructures with adjustable parameters [66,67]. For instance, porous Pt and Pd nanoballs were synthesized by exposing to γ-rays quaternary systems formed with water containing tetraamineplatinum(II), Pt(NH_3_)_4_Cl_2_, or tetraaminepalladium(II), Pd(NH_3_)_4_Cl_2_, complexes, CTAB as a surfactant, pentanol as a co-surfactant, and hexane [68,69]. Bimetallic Pt-Pd nanoparticles with controlled porosity were obtained using hexagonal mesophases with different sizes [70]. Pd nanowires were also synthesized in hexagonal mesophases using electron beams [71]. A combination of chemical oxidation and radiolysis was successfully used to synthesize well-defined 1D Pd-PANI nanocomposites in the confinement of swollen liquid crystalline mesophases [72]. Figure 8 presents TEM images of Pd and Pt-Pd nanostructures synthesized in hexagonal mesophases using ionizing radiation.

The high penetration of ionizing radiation allows the reduction of metallic ions in heterogeneous media but also in porous materials such as zeolites (Figure 7). As the porous medium is filled with a solution containing the metallic ions, the reduction of the ions occurs directly inside the pores; then, the diffusion and coalescence of the atoms and clusters are restricted inside the pores, leading to in situ synthesis of small nanoparticles. Copper [74], silver [75,76] platinum [77,78], and palladium [79] clusters were synthesized by radiolysis in zeolites.

For some applications, it is helpful to have nanoparticles deposited on solid supports. Various supports, such as polymers, oxides, or carbon-based materials, have been used to generate supported metallic nanostructures by ionizing radiation (Figure 7). Common oxides include metallic oxides, such as titanium dioxide (TiO_2_) [80,81,82], alumina (Al_2_O_3_) [83,84], silica (SiO_2_) [85,86], and zinc oxide (ZnO) [87], as well as reduced graphene oxide (rGO) [88,89,90]. Carbon nanotubes also constitute a support of choice [91,92,93]. The metallic nanoparticles can be synthesized in solution and then transferred on a support by impregnation techniques, but it is also possible to perform radiolytic synthesis in the presence of the support. In that case, ions and atoms interact with the surface to form adsorbed clusters which are stabilized by the interactions with the surface, without the addition of a stabilizer. 

## 4. Properties and Applications of the Generated Nanostructures

Radiolysis is a powerful method to synthesize mono- or bi-metallic nanostructures with good control of their size and shape. Pulsed radiolysis also allows the study of the first steps of reduction and nucleation bringing information on the properties of the initial clusters [94]. Thus, the contribution of radiolysis to the investigation of the redox, optical, catalytic, photocatalytic, and electrocatalytic properties of nanoparticles has been considerable [36,94]. Moreover, the radiolytic synthesis of nanomaterials in solution does not require the addition of potentially toxic chemical reducing agents, hence no need for purification, and the irradiation induces the sterilization of the solution, which is very convenient for biomedical applications. Here, we provide examples of the properties and applications of the MNS produced using ionizing radiation. 

### 4.1. Redox Properties and Silver-Based Photography

As already mentioned, the redox properties of metal atoms and clusters differ from those of the bulk and depend on the cluster nuclearity. The pioneering works of Belloni’s group on the reduction of silver ions and nucleation of silver aggregates allowed for an explanation of the principle of the silver photographic development processes [95]. They also elaborated a method to increase the sensitivity of silver photographic films based on the doping with formate ions of the silver halogenide emulsions [96]. 

### 4.2. Optical Properties and Applications

At the nanometric scale, new properties appear, particularly optical properties, which depend on the metal and change with the size and shape of the nanoparticles. These optical properties originate from the localized surface plasmon resonance (LSPR). LSPR corresponds to the collective oscillations of free electrons (conduction band electrons) in MNS induced by the electromagnetic field of an incident light shone upon the MNS. When the light frequency is equal to the resonance frequency of the electronic oscillating system, the light is strongly absorbed by the MNS which then display a characteristic LSPR absorption band. For instance, Ag and Au spherical nanoparticles present a strong LSPR band in the visible spectral domain, around 400 nm and 520 nm, respectively (Figure 9a).

The first controlled synthesis of bimetallic nanoparticles, especially Ag-Au, carried out by radiolysis, enabled the study of their optical properties. It was shown that the absorption band changes with the composition and structures of the nanoparticles (core-shell or alloy) [37]. Anisotropic MNS have also attracted attention due to their optical properties. For example, Au nanorods present two LSPR absorption bands, one around 520 nm called the transverse band and another at higher wavelengths called the longitudinal band. The position of this longitudinal band depends on the aspect ratio: the higher the aspect ratio (i.e., the longer the NR), the more the band is shifted to the red (higher wavelengths) (Figure 9b). In particular, strong absorption in the near-infrared domain is interesting for bio-medical applications.

As the optical properties of MNS are linked to a surface phenomenon, they are affected by the environment of the MNS and may serve as sensors for local changes in the medium: temperature, pH, etc. For example, the shift of the longitudinal LSPR band was used to detect the presence of non-absorbing molecules on their surface [97]. The linear spectral changes of the LSPR of silver nanoparticles toward dopamine concentrations were used for the estimation of dopamine [98]. Reference [99] provides different examples of sensors based on the LSPR properties of gold and silver nanoparticles synthesized using gamma radiation.

MNS also exhibit nonlinear optical properties. For instance, the nonlinear refractive index of colloidal silver nanoparticles stabilized by poly(4-vinylpyridine) in ethylene glycol was determined to be 8.96 × 10^−11^ esu [100]. The nonlinear optical response of gold nanoparticles prepared by γ-radiolysis in water and stabilized by PVA was studied and shown to be size-dependent [101]. From the behavior depending on the excitation fluence, it was suggested that two types of scattering centers were responsible for the optical limitation [102]. A strong enhancement of the second harmonic response with respect to the aspect ratio of gold nanorods synthesized using γ-radiolysis, together with a relative increase in the dipolar hyperpolarizability, was reported [103].

In addition to the optical properties, the resulting thermal properties are of interest, too. Indeed, LSPR phenomena are accompanied by physical effects: optical near-field enhancement, heat generation, and excitation of hot electrons. Thus, plasmonic MNS can behave as nanosources of heat by converting the absorbed photon energy into thermal energy transferred to the surrounding medium. Such an effect is important for chemistry, in particular heterogeneous catalysis, and it was called “plasmon-assisted catalysis” [104]. This effect was also evidenced using Cu nanoparticles stabilized by a thermo-responsive polymer (poly(N-isopropylacrylamide, PNIPAM) [105]. It is thus possible to envisage materials or devices whose functionality would be activated and controlled solely by light.

### 4.3. Applications in Chemistry

#### 4.3.1. Catalysis

In heterogeneous catalysis, chemical reactions in the liquid or gas phase occur at the surface of solid catalysts. The larger the reaction surface, the more efficient the catalysis. Thus, decreasing the nanoparticle size increases the specific surface area (surface area per unit volume) and the number of low-coordination sites (often more active for adsorptions or catalytic processes); thus, it is generally possible to increase activity while limiting the quantity of material and thus reducing costs.

Using ionizing radiation enables the controlled synthesis of very small and well-dispersed nanoparticles on various supports. Due to its high reducing power, the solvated electron can effectively reduce at room temperature metallic ions which are difficult to reduce by chemical methods, such as nickel. For instance, Ni nanoparticles on titanium dioxide (Ni/TiO_2_) were synthesized by radiolysis and used as catalysts for the hydrogenation of benzene [81]. These γ-synthesized catalysts were shown to have a better catalytic efficiency than conventionally obtained catalysts. In particular, it was found that the radiolytic synthesis allows a better reduction of Ni and a better distribution of the nanoparticles on TiO_2_ compared to the conventional synthesis by H_2_ reduction. The higher activity of the γ-synthesized catalysts was attributed to the presence of intermetallic Ni-Ti (Ni_2.66_Ti_1.33_ and Ni_3_Ti) evidenced by X-ray diffraction analysis while the oxidized phase Ni_5_TiO_7_ where Ni is in strong interaction with TiO_2_ is dominant in catalysts obtained by conventional H_2_ reduction and calcination. Ali et al. also showed that Cu NPs synthesized by γ-radiolysis exhibit a better catalytic activity for the reduction of nitrophenol to aminophenol in the presence of NaBH_4_, compared to Cu NPs synthesized by chemical methods [106]. Here, the difference was also explained by a better reduction of Cu and a smaller size of the obtained NPs when using ionizing radiation. Two SiO_2_-supported Ag catalysts with 5 wt% Ag were prepared by impregnation and radiolysis methods and tested for decomposition of N_2_O; it was observed that the catalyst prepared by radiolysis, yielding metallic Ag nanoparticles directly without any reduction treatment, gives a higher conversion and better N_2_ selectivity [107].

Palladium plays a crucial role in catalysis and is involved in various reactions, especially for the formation of C-C bonds in organic reactions such as Heck, Suzuki, and Stille coupling and for the hydrogenation of polyunsaturated hydrocarbons. For instance, the Pd catalysts supported on alumina prepared by the radiolysis method were found to be more active and stable towards the hydrodechlorination of chloro-benzene compared to the catalyst prepared by a conventional method [108]. Pd-urchin-like structures synthesized by a slow radiolytic reduction of Pd(II) in a cetylpyridinium chloride (CPCl) micellar solution have shown very interesting cycling sorption properties for hydrogen storage [109]. Also, the immobilization of nanometals on a solid matrix, such as fabrics, offers an attractive catalytic system because of the convenient separation from the reaction mixture which is highly desirable for many chemical reactions. Ethylenediamine functionalized fabric was used as a support for palladium nanoparticles prepared by gamma radiation. Pd(0) was shown to be simultaneously formed and stabilized through the diamino ligands on the fabric support and was very active for the reduction of 4-nitrophenol [110]. Pd nanoflowers synthesized by radiolysis were used as catalysts in Suzuki–Miyaura C-C coupling reactions. As they display absorption in the visible range, it was shown that their catalytic activity is much enhanced under visible irradiation due to the plasmonic excitation (Figure 10) [111]. 

Lately, Pd NPs synthesized by radiolysis were used as catalysts in Suzuki–Miyaura C-C coupling reactions for intracellular selective modifications of proteins on human thyroglobulin [112]. This first example of a chemoselective change in a native protein using MNS opens a new therapeutic field.

Bimetallic nanoparticles also present interesting catalytic properties which differ from those of their mono-metallic counterparts due to synergetic effects between the two metals. An increase in the activity, selectivity, and durability of the catalysts can be obtained. For instance, the selective hydrogenation of hydrocarbons was performed using Pd-Au and Pd-Ag nanoparticles of controlled composition and structure synthesized using ionizing radiation [83]. The radiolytic synthesis of bimetallic Pt-Au nanoparticles with controlled structures enabled the study of the electronic effects involved in the oxidation of carbon monoxide CO into carbon dioxide CO_2_ and an explanation of the increase in the catalytic activity by associating platinum to gold [113]. Recently, bimetallic Cu–Ni systems supported on γ-alumina catalysts were synthesized using γ-radiation instead of conventional calcination and reduction and tested for the vapor phase selective hydrogenation of levulinic acid to γ-valerolactone. The γ-irradiated catalysts demonstrated a higher hydrogenation activity and stability than conventionally synthesized catalysts; the high activity was attributed to high copper dispersion and well-interactedCu–Ni metallic species [114].

#### 4.3.2. Electrocatalysis

Pt-based nanoparticles are very efficient in most electrocatalytic reactions involved in fuel cells. Pt-based catalysts prepared by radiolysis in different conditions and supported on graphite powder were very active for the hydrogen evolution reaction (HER) with an activity higher than that of the commercial compound (Pt) Johnson Matthey. Pt clusters (synthesized under a CO atmosphere) showed the highest activity. This enhanced activity appears to be linked essentially with two main parameters: the mean nanoparticle size and, for comparable sizes, the surface morphology of the materials. The results and the stability of the electrodes suggested that the small particle sizes and the good dispersity on the carbon support were maintained during the HER [115]. 

However, poisoning of platinum surface by CO molecules limits the lifetime of electrocatalysts. The combination of platinum with another metal such as ruthenium or gold avoids (or restrains) this poisoning and increases the lifetime of the catalyst. Promising results were obtained for the oxidation of methanol using very small bi- and tri-metallic nanoparticles, Pt-Au, Pt-Ru, and Pt-Ru-Sn, of a few nanometers, synthesized by radiolysis, and deposited on Vulcan carbon [22]. Pt nanodendrites prepared by 2 MeV electron beam radiolysis in a PVP micelle solution were also shown to display drastically enhanced electrochemical catalytic performances toward methanol oxidation, and to possess a better CO tolerance for methanol oxidation than commercial catalysts [116].

Palladium is more abundant than platinum. Pd-based nanomaterials synthesized using ionizing radiation (Pd on carbon nanotubes, Pd-Au nanoparticles) were found to be very active for the oxidation of ethanol in alkaline media [91]. The homogeneous coating of carbon nanotubes with 2 nm-Pd nanoparticles was achieved using the supramolecular auto-organization of amphiphilic molecules as a template. The resulting Pd nanoparticles/carbon nanotube nanohybrids synthesized by electron beams showed superior activity in ethanol oxidation compared to analogous systems, and very high current densities were obtained [91]. Lately, it was shown that Pd nanoparticles supported on carbon nanotubes synthesized by γ-radiolysis were also promising electrocatalysts for oxygen reduction reactions [117]. Pd nanowires and porous nanoballs synthesized (respectively by gamma-rays and electron beams) in mesophases exhibit very high activity for ethanol oxidation in alkaline media [69,71]. The Pd_shell_−Au_core_ nanostructures synthesized in mesophases were found to be promising for application in direct ethanol fuel cells as they exhibit a very good electrocatalytic activity and high stability (Figure 11) [118].

Co- and Ni-based particles supported on carbon were synthesized by gamma irradiation and used to fabricate electrodes for full cells. The obtained electrodes showed high catalytic activity for the oxygen reduction reaction [119].

Using ionizing radiation also enabled the one-pot synthesis of Au-based nanoparticles on reduced graphene oxide rGO with the simultaneous reduction of the metallic ions and graphene oxide used as a support. The formed nanocomposites showed a high activity for the electro-oxidation of glucose [120]. The electrocatalytic oxidation of glycerol by metal electrocatalysts is an effective method for low-energy input hydrogen production in membrane reactors in alkaline conditions. Au-based nanoparticles (Au and Au-Ag NP) were directly synthesized on (and in) carbon paper fibers for application in glycerol electro-oxidation [121]. 

#### 4.3.3. Photocatalysis

Photocatalysis is a process in which light energy is used to initiate chemical reactions; it can be used in advanced oxidation processes for air or water depollution, self-cleaning surfaces, hydrogen generation, or carbon dioxide reduction. Photocatalysts are materials that possess a catalytic activity when exposed to light. The most common photocatalysts are transition metal oxides and semiconductors, in particular titanium dioxide, TiO_2_. TiO_2_ is a very efficient photocatalyst due to its strong oxidation capacity, high photochemical and biological stability, and low cost. However, TiO_2_ shows some limitations: first, the quantum yield is low due to fast charge carriers (electron/hole) recombination, and second, its activation occurs under UV irradiation because of the value of its band gap depending on the crystalline structure (3.2 eV for anatase and 3.0 eV for rutile). These limitations can be overcome by doping TiO_2_ with plasmonic metallic nanoparticles synthesized by radiolysis. The plasmonic nanoparticles increase the photocatalytic activity of TiO_2_: (i) in the UV domain by scavenging the electrons and so diminishing the electron-hole recombination; (ii) in the visible domain by their LSPR absorption [122]. For instance, it was reported that TiO_2_ modified with Ag nanoparticles using gamma irradiation presents a higher photocatalytic activity compared to TiO_2_ for the degradation of organic pollutants [123,124]. It was also shown that the surface modification of TiO_2_ with both Ag nanoparticles and CuO nanoclusters synthesized by radiolysis induces an increase in the photocatalytic activity under both UV and visible light for depollution [125]. Radiolysis of aqueous solutions containing metallic ions in the presence of TiO_2_ leads to the formation of small mono- or bi-metallic nanoparticles well-dispersed on the surface of the photocatalyst. Several studies reported beneficial synergetic effects between the two metals in the catalytic activity of the nanocomposites for H_2_ generation by using bimetallic nanoparticles such as Au-Ni [82], Ni-Pd [126], and Pt-Ni [127]. Conjugated polymer nanostructures emerge as new photocatalysts that are very active under visible light [128]. Their surface modification with metal (Pt, Pt-Ni) nanoparticles induced by radiolysis has led to very active photocatalysts for hydrogen generation [129]. Recent research works concern the development of co-catalysts without noble metals and based on abundant elements (Ni, Fe, Cu…).

### 4.4. Biomedical Applications

In recent decades, with the development of nanoscience, many biocompatible metallic nanoparticles have been synthesized and studied for therapeutic and diagnostic purposes or applications in the field of sensors and bio-imaging [130,131]. In particular, gold NPs have appeared as good contrast agents in microscopy and X-ray imaging, as well as magnetic resonance imaging. The use of NPs for simultaneous diagnostic and therapy (theragnostic) also opens the route toward personalized medicine of tomorrow. The main advantage of the MNS synthesized in solution using ionizing radiation is that the solutions are sterile.

#### 4.4.1. Antimicrobial Properties

Some metals (Cu, Ag, Au, Pt) in the form of nanoparticles have antimicrobial properties [132]. It has been demonstrated that nanocomposite hydrogels containing Ag nanoparticles synthesized by γ-irradiation present an effective antimicrobial activity [44,49,133,134]. Ag NPs stabilized by PVA and synthesized by radiolysis possess a higher antibacterial activity than that of Ag NPs synthesized by other methods [45]. The use of silver nanoparticles formed by gamma and electron beam irradiations on titanium dioxide as a coating on polyurethane catheters was also studied for antimicrobial activity [135]. It is reported that a TiO_2_ surface modified by Ag@CuO clusters synthesized by γ-radiolysis inhibits fungal growth [136]. Textile fabrics made of natural or synthetic fibers were used as support materials for stabilizing Ag nanoparticles induced by high-energy electron beams. The Ag nanoparticles immobilized on the support textile fabric exhibited an excellent antibacterial activity across a wide antibacterial spectrum, even after a durability test involving washing the fabric 100 times [137]. 

#### 4.4.2. Radiotherapy and Hadrontherapy

Since the first report of J.F. Hainfield et al. in 2004 [138], the association of metallic nanoparticles with X-rays (radiotherapy) or particle beams (hadrontherapy) seems a promising approach to improve treatments in cancer therapy [139,140,141]. Indeed, metallic NPs with a high atomic number Z (gold and platinum) combined with ionizing radiation can enhance the effect of radiotherapy or hadrontherapy. Research has focused, in particular, on the possibility to improve their therapeutic effects and tumor targeting by combining them with co-agents or by adsorbing active molecules on their surface. 

For instance, it was shown that Pt nanoflowers stabilized by PEG-diamine and synthesized by γ-radiolysis are non-toxic and enable a substantial effect on cancer cells [142]. Au-Pt NPs stabilized by polyethylene glycol (PEG) and synthesized by radiolysis amplify the effect of radiation, in particular, the induction of nanoscale damage (>2 nm) [143]. It was also demonstrated that Pt nanoparticles combined with ionizing radiation can kill radioresistant bacteria [144]. 

Moreover, in radiation therapy it is also very important to control the radiation dose delivered to the patient. Recently, it was reported that the formation of plasmonic gold nanoparticles from gold salts in nanocomposite gels under radiation could be used as a simple dosimeter for the detection of therapeutic levels of radiation [145,146]. 

#### 4.4.3. Hyperthermal Therapy

In cancer treatments, gold nanoparticles can also be used to kill tumors by photothermal therapy. Indeed, human cells have the ability of spontaneous self-destruction when subjected to temperatures above the body temperature (37 °C) by six or eight degrees. Cancer cells are even more sensitive to such a rise in temperature. Under light illumination, plasmonic MNS can behave as a nanosource of heat [147]. Thus, gold nanoparticles in contact with cancer cells and excited by a laser in their LSPR band heat the medium and may destroy the cells. Tuning the laser intensity allows for the destruction of only the cancer cells without damaging the surrounding healthy cells. As the spectral window of biological tissues ranges from 650 to 900 nm, anisotropic Au MNS (nanorods, bipyramids, nanostars) can be used since they exhibit a second LSPR band in the near-infrared domain [64]; targeting a solid tumor (skin cancer, for example), and then its regression can be induced by light irradiation.

## 5. Conclusions and Perspectives

The contribution of ionizing radiation to the synthesis and the study of MNS properties, which vary with size and shape, is significant. Radiolysis is a powerful technique to synthesize (without the addition of chemical reducing agents) mono- and multi-metallic nanoparticles of controlled size, shape, composition, and structure in solution, in heterogeneous media, and on supports. The formed MNS can have applications in various domains, such as catalysis, electrocatalysis, photocatalysis, and nanomedicine. Industries with the ability to scale up can use this technique, as large volumes can be irradiated by industrial irradiation facilities worldwide. Thus, protocols for the radiolytic synthesis of nanoparticle-based materials (for example, electrocatalytic materials) can be transposed to industrial production. Finally, it must be mentioned that ionizing radiation can be used for synthesizing nanoparticles other than MNS: nanostructured polymers, semiconductors, quantum dots (ZnS, CdS, PbS), and hybrid nanomaterials for many applications in numerous domains.

## Figures and Tables

**Figure 1 materials-17-00364-f001:**
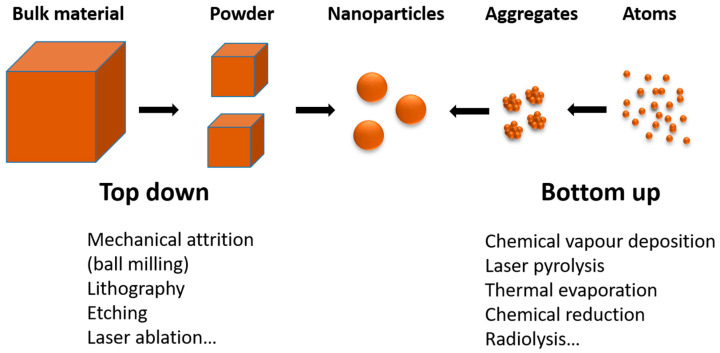
Scheme illustrating top-down and bottom-up approaches for the synthesis of nanostructures.

**Figure 2 materials-17-00364-f002:**
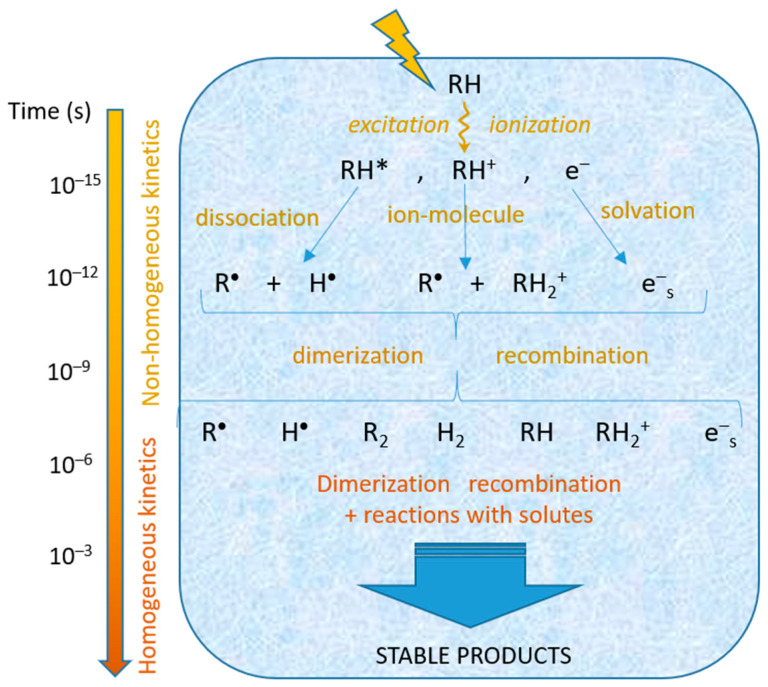
Scheme of the radiolysis of a protic solvent RH showing the reactive species generated by ionizing radiation.

**Figure 3 materials-17-00364-f003:**
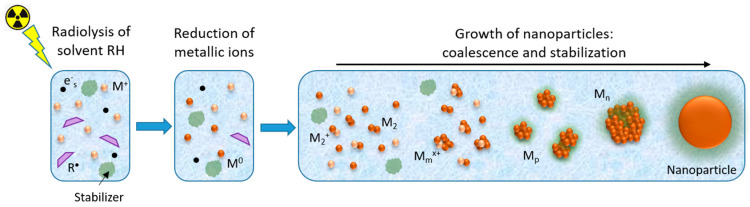
Scheme depicting the main processes in the synthesis of metallic nanoparticles using ionizing radiation. The ionizing radiation interacts with the solvent molecules, RH, generating reducing species, solvated electrons (e^−^_s_,) and radicals (R^•^); these species react with the metallic ions M^+^ to give the atoms M. The metallic ions and atoms further coalesce to form clusters and then nanoparticles, which are stabilized by ligands, polymers, surfactants, or supports already present in the solution.

**Figure 4 materials-17-00364-f004:**
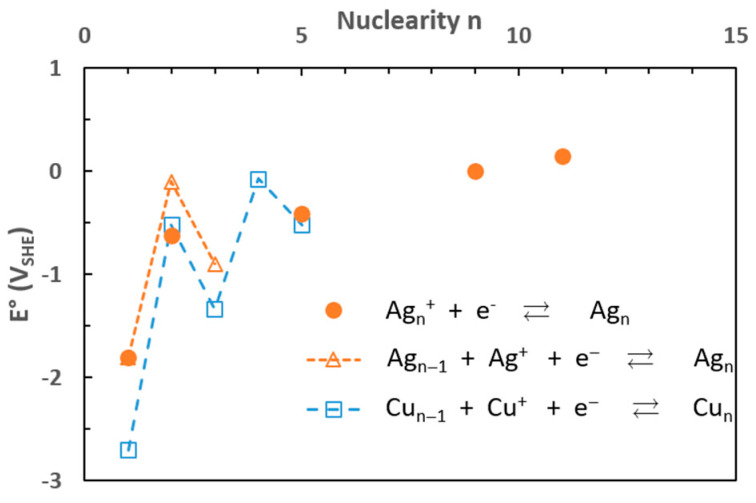
Redox potential of metallic aggregates as a function of the nuclearity n of the aggregates: ● (Ag_n_^+^/Ag_n_) (data from [32]); Δ (Ag^+^/Ag_n_) and □ (Cu^+^/Cu_n_) (data from [33]).

**Figure 5 materials-17-00364-f005:**
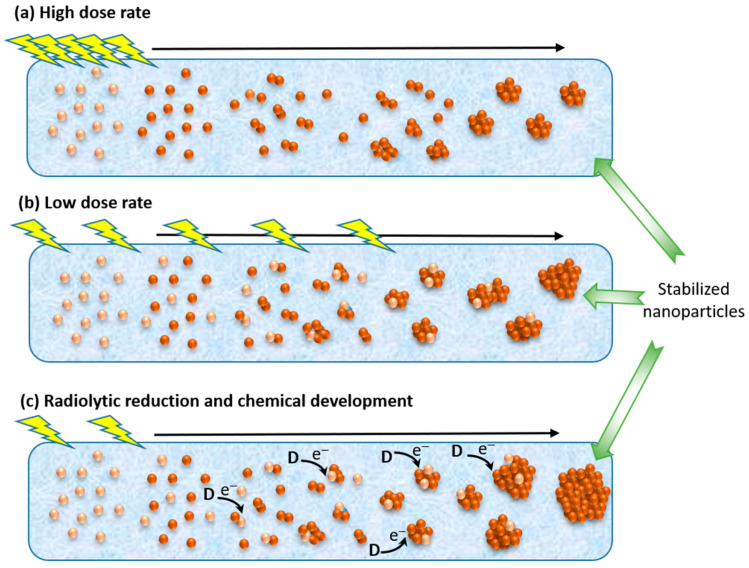
Influence of the dose rate (high (**a**) and low (**b**)) and the presence of a reductant D (**c**) on the growth of metallic nanoparticles.

**Figure 6 materials-17-00364-f006:**
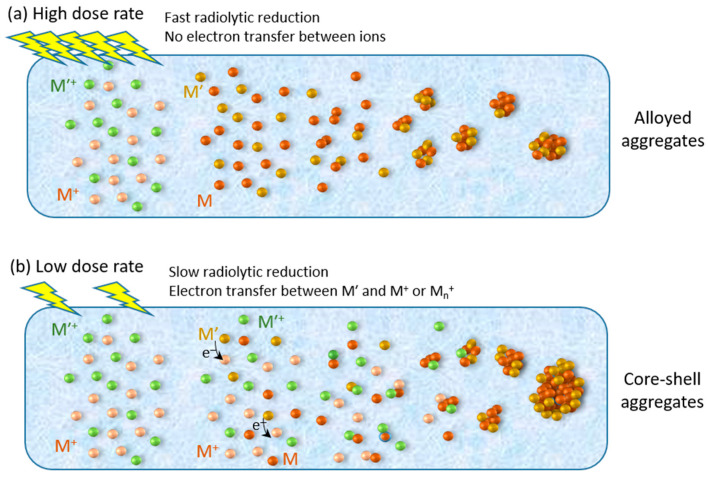
Influence of the dose rate (high (**a**), low (**b**)) on the structure of bimetallic nanoparticles.

**Figure 7 materials-17-00364-f007:**
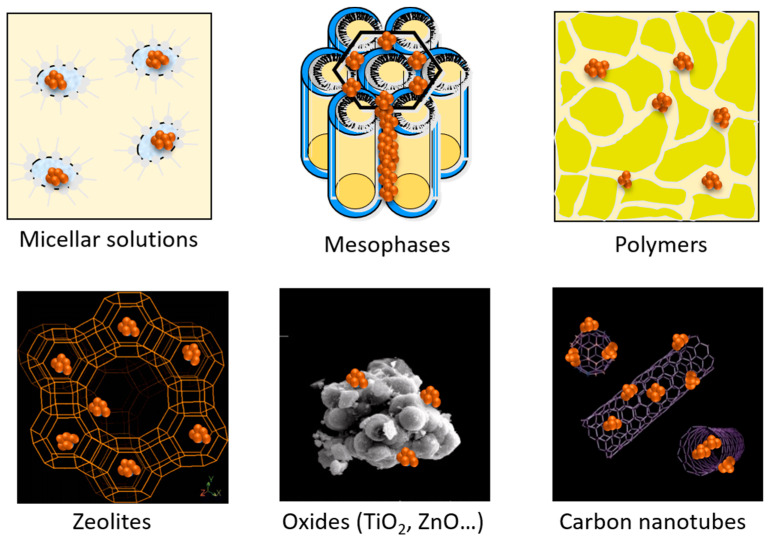
Scheme of aggregates produced by ionizing radiation in different soft templates (micelles, mesophases, polymers) and hard matrices (zeolites, oxides, carbon materials).

**Figure 8 materials-17-00364-f008:**
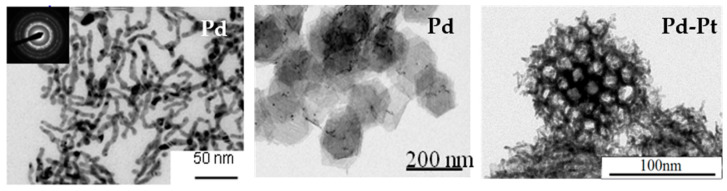
TEM images of Pd nanowires [71], Pd nanosheets [73], and Pd-Pt nanoballs [70] synthesized in hexagonal mesophases using ionizing radiation.

**Figure 9 materials-17-00364-f009:**
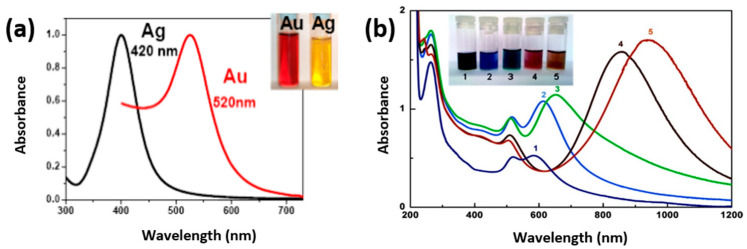
UV–Visible absorption spectra of (**a**) spherical Au and Ag nanoparticles and (**b**) Au nanorods synthesized by γ-radiolysis with different aspect ratios from 1.7 (curve 1) up to 5.2 (curve 5) Inset: image of the vials containing the solutions. ((**b**) reprinted with permission from Ref. [64]. Copyright 2010 American Chemical Society).

**Figure 10 materials-17-00364-f010:**
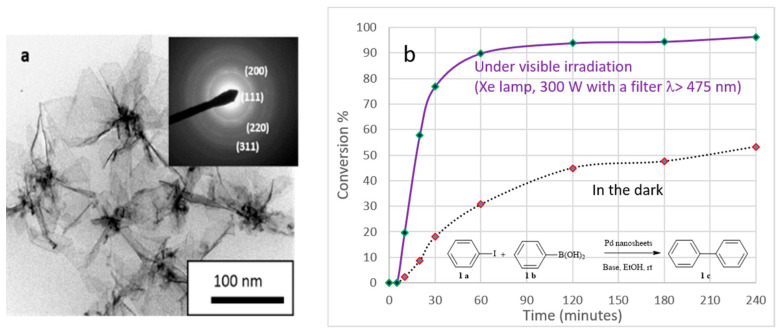
(**a**) TEM image of Pd nanoflowers synthesized by a radiolytic reduction of Pd^II^(acac)_2_ in ethanol under a CO atmosphere; (**b**) Catalytic activity of the formed palladium nanoflowers in the dark and under visible light irradiation for the Suzuki–Miyaura coupling reaction between iodobenzene (1a) and phenyl boric acid (1b) (scheme given in inset) (reproduced from Ref. [111] with permission from the Centre National de la Recherche Scientifique (CNRS) and the Royal Society of Chemistry).

**Figure 11 materials-17-00364-f011:**
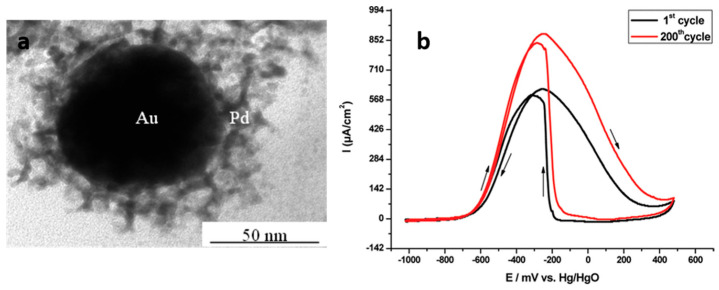
(**a**) TEM image of a Pd_shell_-Au_core_ nanostructure formed in hexagonal mesophases; (**b**) cyclic voltammograms associated with the electrocatalytic oxidation of ethanol in alkaline medium showing the high activity of the Pd_shell_-Au_core_ nanstructures (reprinted with permission from Ref. [118]. Copyright 2009 American Chemical Society).

**Table 1 materials-17-00364-t001:** Radiolytic yields, G, of the primary species for water radiolysis under γ-rays (LET of 0.23 eV/nm) and under 12 MeV alpha particles (LET of 108 eV/nm) in deoxygenated conditions [21].

G-Values (µmol/J)	^•^OH	H^•^	e^−^_aq_	H_3_O^+^	H_2_O_2_	H_2_
γ-rays (LET = 0.23 EV/nm)	0.28	0.062	0.28	0.28	0.073	0.047
α particles (LET of 108 eV/nm)	0.056	0.028	0.044	0.044	0.11	0.11

**Table 2 materials-17-00364-t002:** Examples of redox couples involving reducing radicals and their reduction potential in water [24].

Ox/Red	Equation	E° (V_SHE_)
CO_2_/^•^COOH	CO_2_ + e^−^ + H^+^ ⇄ ^•^COOH	−1.82
CH_2_O/H_2_C^•^H(OH)	CH_2_O + e^−^ + H^+^ ⇄ H_2_C^•^H(OH)	−1.18
CH_3_CHO/CH_3_C^•^H(OH)	CH_3_CHO + e^−^ + H^+^ ⇄ CH_3_C^•^H(OH)	−1.25
(CH_3_)_2_CO/(CH_3_)_2_C^•^OH	(CH_3_)_2_CO + e^−^ + H^+^ ⇄ (CH_3_)_2_C^•^OH	−1.39

**Table 3 materials-17-00364-t003:** Influence of ligands on the redox potential of the (Ag^+^/Ag) couple in aqueous solution.

Ox/Red	Ag^+^_(aq)_/Ag^0^_(aq)_	Ag^I^(CN)_2_^−^/Ag^0^(CN)_2_^2−^	Ag^I^(NH_3_)_2_^+^/ Ag^0^(NH_3_)_2_	Ag^I^(EDTA)^3−^/Ag^0^(EDTA)^4−^	Ag^I^(Cl)_4_^3−^/Ag^0^(Cl)_4_^4−^
**E° (V_SHE_)**	−1.8 [28]	−2.5 [29]	−2.4 [30]	−2.2 [29]	−2.35 [31]

## Data Availability

Not applicable as no new data were created or analyzed in this paper.

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
