# Peer review of "Synthesis of Metallic Nanostructures Using Ionizing Radiation and Their Applications"

_materials, 2024, doi:10.3390/ma17020364_

Round 1

Reviewer 1 Report

Comments and Suggestions for Authors

This paper is very well organized and written review of the "Radiolysis method" that is proven to be effective in producing metallic nanoparticles with various form factor, composition and structures.  No complains to make about the content.  One optional change to suggest is to include the discussion on "the limiting conditions" of the method.  For example, since formation of nanoparticles in solvent would increase the radiation absorbance of total solution and limit the penetration of depth of radiation dose.  Control on the precursor concentration or application of agitation to increase uniformity in exposure to radiation may be necessary.  The point is that they may exist various type of limiting situations and mentioning of them can make this review paper to be more comprehensive and complete.  

Comments on the Quality of English Language

The text of this paper is very well written and is almost perfect except for a few occasional errors or missing definitions.  Examples for each are:

Page 3 line 118: "according of" should be changed to "according to" 

VNHE definition: NHE seem referring to "normal hydrogen electrode" yet its definition is missing.  NHE is rarely used so it may need to be defined, or use SHE that is more familiar term for the non-expert.  

There are a few places like above that are in need of corrections.  

Author Response

We thank the referee for the positive comments and suggestions.

As suggested, we have added a small paragraph dealing with the limiting conditions of the radiolytic method at the end of section 2.1 page 5

Comments on the Quality of English Language

The text of this paper is very well written and is almost perfect except for a few occasional errors or missing definitions.  Examples for each are:

Page 3 line 118: "according of" should be changed to "according to" 

This correction have been done

VNHE definition: NHE seem referring to "normal hydrogen electrode" yet its definition is missing.  NHE is rarely used so it may need to be defined, or use SHE that is more familiar term for the non-expert.  

We have changed NHE to SHE as suggested.

There are a few places like above that are in need of corrections.

Several typographical mistakes have been corrected, we hope all.

Reviewer 2 Report

Comments and Suggestions for Authors

Paper gives an overview of the use of ionizing radiations for the synthesis of metallic nanoparticles. Will be of interest for researches since includes from basic definitions to applications. The area is of great interest since it is descrived as clean alternative for the preparation of nanoparticles with variety applications. Article is informative and is properly written. Should be accepted after minor revision. 

Considering the general title of "metallic nps", a sumary table is suggested, describing the most used metal ions within this field and particle size achieved, morphology, radiation used, application, etc. 

Table 3 parentesis missing for Ag0(NH3 and a minnus sing I think for Ag0(CN)22. Check prentesis for Ag0(aq). instead of notation 1 Ref. 28; 2 Ref . 29; 3 Ref. 30; 4 Ref. 31 direct citation is recomended. 

Figure 4 nuclearity is misspelled. 

chapter 3.4.3 should be revised. includes only two paragraphs, being the first one descriptive and basic. The second one mentions catalytic application, bein these sections descrived earlier (3.3). 

Author Response

We thank the referee for the positive comments and suggestions. However, we have not included a summary table as it would have been too large and not necessarily easy to read with many references for a wide range of mono- and bi-metallic nanostructures, supported or not, and various applications.

Table 3 parentesis missing for Ag0(NH3 and a minnus sing I think for Ag0(CN)22. Check prentesis for Ag0(aq). instead of notation 1 Ref. 28; 2 Ref . 29; 3 Ref. 30; 4 Ref. 31 direct citation is recomended. 

The table 3 has been corrected and modified as suggested.

Figure 4 nuclearity is misspelled. 

The axis title has been corrected.

chapter 3.4.3 should be revised. includes only two paragraphs, being the first one descriptive and basic. The second one mentions catalytic application, bein these sections descrived earlier (3.3). 

Chapter 3.4.3 has been removed; the first paragraph has been integrated into the introduction of section 3.4, and the mentioned catalytic application transferred to section 3.3.1.

Reviewer 3 Report

Comments and Suggestions for Authors

The manuscript by Remita and Lampre, titled “Synthesis of metallic nanostructures by ionizing radiations and their applications” overviews the preparation of metal nanoparticles by ionizing radiation. The English is good, literature is well quoted, and the topic is interesting. Therefore, I suggest publication in Materials after minor revision.

There are some typos throughout the text (such as the title of figure 4 and the syntax of the sentence in rows 68-69).

Comments on the Quality of English Language

Minor editing required

Author Response

We thank the referee for the positive comments. 

There are some typos throughout the text (such as the title of figure 4 and the syntax of the sentence in rows 68-69).

Several typographical mistakes have been corrected throughout the text, we hope all. The syntax of the sentence has been changed.